# Evaluation of the performance and stability of early maturing orange-fleshed sweetpotato genotypes in selected areas in Ethiopia

Bililign Mekonnen[1]*, Fekadu Gurmu[2]

1 Sidama Region Agricultural Research Institute (SIRARI), Hawassa Agricultural Research Centre, Hawassa, Ethiopia, 2 Ethiopian Institute of Agricultural Research (EIAR), Addis Ababa, Ethiopia

* bililign.m@gmail.com

**Data Availability Statement:** All relevant data are within the manuscript and its Supporting Information files.

**Funding:** This work was supported by the Ethiopian Institute of Agricultural Research (EIAR)

## Abstract

Orange-fleshed sweetpotato varieties that mature and harvest sooner play an important role in addressing food and nutrition demands in areas where irregular rainfall makes sustainable production challenging. A national variety trial was conducted in 2021 and 2022 during the main cropping season using ten OFSP genotypes in three locations in Sidama, South, and Oromia regions of Ethiopia, namely Hawassa, Arbaminch, and Koka, respectively. The objective of this study was to develop and select early-maturing and high-yielding OFSP genotypes for short-term harvesting (3–4 months). The field trial was conducted in a randomized complete block design with three replications. Data were collected on root yield and yield-related traits, sweetpotato virus disease reactions (SPVD), root dry matter (DMC), and beta-carotene contents (BCC) and were subjected to analysis of variance. A genotype plus genotype by environment interaction (GGE) bi-plot was also used to determine genotype stability. The results showed the presence of highly significant (p<0.001) differences among locations and genotypes, reflecting the existence of differential responses among genotypes in varied locations. Based on combined analysis, G3 (13NC9350A-9-3) outperformed the other genotypes for most of the traits considered, except for DMC, i.e., which has an equivalent to the check variety (Alamura) and showed a yield advantage of 41.4% over it. The GGE biplot also revealed that the G3 was the vertex genotype with the consistent performance in all environments. It had a low score of 1.39 on the 1–9 scoring scale, indicating that it falls within the resistant range, with adequate levels of BCC (5.5 mg/100 g) and DMC (30.0%). Furthermore, G3 is an early-maturing variety, which allows other crops to be grown in double and relay cropping systems. Therefore, based on its outstanding performance, G3 is recommended for verification and release for cultivation in mid- to low-land areas in Ethiopia.

## Introduction

Sweetpotato [*Ipomoea batatas* (L) Lam., 2n = 6x = 90] is an important root crop that can thrive in a wide range of agro-ecological conditions, including areas with erratic rainfall. It can

and sub-grants from SweetGAINS project of CIP (INV-002961), funded by the Bill and Melinda Gates Foundation. The funders had no role in study design, data collection and analysis, decision to publish, or preparation of the manuscript.

**Competing interests:** The authors have declared that no competing interests exist.

significantly contribute to the global food system by addressing the food and nutrition demands of communities with limited resources, particularly in developing countries [1]. The orange-fleshed sweetpotato type has an advantage over the white sweetpotato type in terms of providing a healthy and nutritious diet in areas where food and nutritional security are major issues [2,3]. Globally, sweetpotato ranks seventh in production, after rice, wheat, maize, potato, barley, and cassava, with an annual yield of around 119 million metric tons [4]. In Africa, it covered 4.21 million hectares of land, producing around 29.11 million tones [4]. In Ethiopia, it was grown on approximately 62,115 hectares in Meher season (in one of the production seasons), with a total production of more than 1.5 million tones [5]. According to the Central Statistical Agency (CSA) report (2020), it is the 2nd most important root crop next to Enset (*Ensete ventricosum*) crop in Ethiopia [6]. More importantly, orange-fleshed sweetpotato (OFSP) varieties, have received a due attention for the intervention as a means for nutrition-sensitive agriculture, and emergency responses in low-land areas, where rainfall is erratic, and for food diversification strategies [7].

Currently, emerging issues such as recurrent drought caused by climate induced changes, vitamin A deficiency (VAD), a rapidly growing human population, and susceptibility of the existing sweetpotato varieties to sweetpotato virus disease and expansion of urbanization urge agricultural researchers to emphasize on the development of nutritious, high-yielding, and early-maturing OFSP varieties for wider consumption and expansion [8–10]. According to Belesova et al. (2019), in agricultural terms, drought is defined as when the amount of moisture in the soil no longer meets the needs of a specific crop in order to achieve the highest yield [11]. Thus, it is also one of the major abiotic factor that hinder crops from obtaining their potential maximum yield, including sweetpotato production. Although sweetpotato is regarded as drought tolerant crop, the onset of drought (early, mid-season or terminal), as well as the intensity and duration, significantly affect how much storage root yield occurs. The performance of genotypes for their performance is crucial different sweetpotato cultivars respond differently to varying soil moisture contents [12].

Another health concern in the current scenarios is micronutrient deficiency, of which vitamin A deficiency is one of the five key micronutrient mineral deficient nutrients. It remains a serious health concern in resource-poor rural communities around the world, including Ethiopia [13,14]. Various strategic approaches, viz., supplementation, fortification, diversification, and biofortification, are being recommended as possible avenues to curb malnutrition. The cultivation and consumption of biofortified crops that are commonly grown in a given area is regarded as one of the most cost-effective solutions for addressing malnutrition and associated problems caused by a lack of essential pigments in commonly consumed foods. In this perspective, biofortified orange-fleshed sweetpotato varieties play an important role in alleviating vitamin A deficiency, as this crop is well known as a readily available source of a nutritious diet [15,16]. The production and productivity of nutritious sweetpotato varieties can ensure the availability of foods for enhanced food system. However, cultivation is hampered by a number of biotic and abiotic constraints, including virus diseases, fungal diseases, insect pests, and weeds; biotic factors such as declining soil fertility and recurring drought; and socioeconomic factors such as a lack of clean planting materials, insufficient availability of high-yielding varieties, etc [17]. Sweetpotato virus disease (SPVD) is one of the most devastating among biotic factors; thus, more than 20 virus diseases are known to infect cultivated sweetpotatoes worldwide [18]. The SPVD, caused by the co-infection of sweetpotato feathery mottle virus (SFMV) and sweetpotato chlorotic stunt virus (SPCSV), is a devastating disease of the sweetpotato crop globally [19,20]. According to Abebe et al. (2023), the degree of yield loss to virus infection varies based on the resistance levels of the sweetpotato genotypes, the crop's growth stage at which the diseases occur, and the agro-ecology of the country [21]. The degree of yield

loss or damage caused by SPVD reported in various regions of the world ranged from 50 to 100%, depending on the severity of the infestation [19,22]. Mekonen et al. (2016) found that synergistic infection with SPVMV and SPCSV reduced yield by up to 37% in Ethiopia [23]. In terms of efficacy, sustainability, environmental friendliness, and low production costs, selecting and implementing resistant or tolerant genotypes is the best choice for managing plant diseases such as SPVD [24,25].

Furthermore, global climate change-induced moisture stress currently poses a risk to sustainable crop production and causing existential food concerns for smallholder farmers [11]. This, along with the other challenges mentioned above, is currently affecting sweetpotato production and productivity in Ethiopia. Sweetpotato varieties that mature early are a better resort for such problems. However, varietal differences exist, particularly in the intensity and commencement of droughts caused by erratic/irregular rainfall [26]. Early-maturing sweetpotato varieties are widely regarded as the best option for addressing the effects of variable or irregular rainfall since they bulk and mature quickly, with harvesting happens within 3–4 months. In Ethiopia, sweetpotato varieties are categorized into three groups based on their maturity time. The first groups are early maturing, maturing within 90–120 days; the second groups are medium maturing, maturing within 121–150 days; and the third groups are late maturing, taking more than 150 days [27].

Early maturing groups are recommended for lowland areas characterized with erratic /irregular rainfall, while medium and late maturing groups are recommended for mid-altitude areas with medium to high rainfall during the growing season [27]. Accordingly, early maturing varieties have an advantage over medium to late maturing varieties in terms of producing food in a short period of time with a limited amount of available rainfall eventually ensuring food availability for needy areas. Although sweetpotatoes are known to be adaptable to a wide range of agro-ecological zones, however, varietal differences exist due to their sensitivity to genotype-environment interaction [28–31]. The authors explained the impact of genotype-by-environment interaction (GEI) on sweetpotato genotypes and their stability in diverse environments across their respective countries using a genotype-by-environment interaction (GGE) biplot analysis, which quantifies the genotype-by-environment interaction (GxE) effect based on multi-location trials [32,33]. Based on the above justification, a national variety trial was conducted over two consecutive years (2021–2022) at three locations (Hawassa, Koka, and Arbaminch) using ten OFSP genotypes (nine selected, and one released variety, which is widely being cultivated in major sweetpotato growing areas), with the objective of identifying early-maturing varieties that can mature early and withstand low moisture stress caused by irregular or erratic rainfall.

## Materials and methods

### Description of study areas and its weather conditions

The field trial was conducted for two consecutive years (2021 and 2022) at three locations namely Hawassa, Arbaminch, Koka in Sidama, South, and Oromia regions, respectively, in Ethiopia. In terms of meteorological conditions during the sweetpotato growing season in 2021, Arbaminch, Hawassa, and Koka areas received 541.2, 993.5, and 523.6 mm of rain, respectively. However, the growing season rainfall was only 291.2, 601 and 470 mm in 2022, in that respective order. This showed that sweetpotato plants received lower amount of rainfall in 2022 than 2021; and the rainfall received during the growing season was the lowest at Arbaminch, intermediate at Koka and the highest at Hawassa (S1 Data). As a result of inconsistent rainfall, three months of low moisture stress were experienced in 2022. In October and September, Arbaminch and Koka received more rainfall than Hawassa did. Furthermore, whether

the year was good or bad, rainfall in August and September was always less than what was needed for crops in Arbaminch, as it was in October and November in Koka. In Hawassa, August and September were occasionally drier (S1 Data).

**Testing sites permission.**   This work is a part of the Ethiopian Institute of Agricultural Research (EIAR). The institute already had linkages with regional agricultural research centers and agricultural universities in Ethiopia to collaborate on any research activities. Furthermore, when designing field experiments, any field research activities must go through a review process that includes testing locations. The activity can be carried out at the designated locations (sites) without any limitations or requests for site authorization once it has successfully completed this process.

## Description of experimental materials

Botanical seeds derived from 16 polycross families were introduced from Uganda in 2016. Prior to establish field trial, over 400 botanical seeds of all families were characterized based on heritable traits such as root yield, root flesh color, i.e., deep orange flesh color, is more favored because deeper orange flesh color is associated with higher beta-carotene content, reaction to disease and insect pests. Based on the traits stated above, genotypes that demonstrated the best performance were promoted to the next yield performance testing, while those having disease signs in each family were discarded. At the advanced stage, nine selected genotypes and one standard check, released in 2019 and were tested in a national variety trial at three locations (Hawassa, Koka, and Arbaminch) from 2021 to 2022 following variety releasing requirements in Ethiopia (S1 Table).

## Field design and trial management

The field trial was established in a Randomized Complete Block Design (RCBD) replicated thrice. Each genotype is planted on a plot size of 2.4 m width and 3 m length (a total plot area of 7.2 m$^2$). Each plot consisted of four rows/ridges, each row accommodated 10 plants/row and a total of 40 plants/cuttings/plot. The spacing between plants and rows were 0.3 m and 0.6 m, respectively. The spacing between replications was 1.5 m. In each plot, one plant (vine cutting) of 30 cm length with 5–8 nodes was planted in the prepared holes by burying one-third of the plant in the soil. After one week, dead plants (vine cuttings) were replaced by new ones (to maintain uniformity of plant population per plot), and after the fourth week, earthening was done, and all plots were manually weeded according to the recommendation for sweetpotato production in Ethiopia [34].

## Data collection

Data were collected on root yield and root-yield related traits on a plot basis from the two middle rows, with the exception of the two plants grown at each ends of the row and the two border rows. Data on SPVD severity was recorded prior to one month of harvesting, whereas data on root yield and yield-related traits were recorded during harvest [35]. Data for root dry matter content (DMC) were determined using the procedures suggested by Tairo et al. (2008) and Carey and Reynoso (1999), with minor modifications. A sample of four roots from each plot was collected [36,37]. For each genotype, samples of 200 g of undamaged roots were sampled from each replication. The prepared samples were put in a paper bag and dried in an oven set to 70˚C for 72 hours, and recording was done until they maintained a constant weight. Finally, the dried samples were weighed using a sensitive balance or weighing scale, and the dry matter content was calculated as a percentage of the fresh weight. Beta-carotene content (BCC) was estimated using a standard color chart [38].

## Statistical analysis

**Analysis of variance.** Data for each location were analyzed separately before performing a combined analysis to test the homogeneity of error variance across locations and over years using the F-ratio, which was found to be non-significant (data not displayed here), then the combined analysis was computed using SAS software version 9.3 [39]. The Fisher's protected least significant difference (LSD) test was performed to compare the treatments at 1% and 5% confidence levels [40].

The following statistical model was used for combined analysis of variance:

$$Y_{ijkl} = \mu + G_i + L_j + Y_k + GL_{ij} + GY_{ik} + LY_{jk} + GLY_{ijk} + B_{l(jk)} + \varepsilon_{ijkl}$$

Where: $Y_{ijkl}$ is observed value of genotype $i$ in block $l$ nested in (location $j$ and year $k$), $\mu$ is grand mean, $G_i$ is genotype effect, $L_j$ is location effect, $Y_k$ is year effect, $GL_{ij}$, $GY_{ik}$ and $GLY_{ijk}$ are the interaction effect of genotype $i$ with location $j$, genotype $i$ with year k and genotype $i$ with location $l$ and year $k$, $_{ijkl}$ is error (residual) effect.

**Genotype by environment interaction (GEI) analysis.** The data were graphically analyzed in GenStat (18th version) for GEI and stability of genotypes using the GGE biplot procedure [41–43]. The following model for the GGE biplot based on singular value decomposition (SVD) of the principal components (PCs) was used:

$$\bar{Y}_{ij} - \mu_i - \beta_j = \sum_{k=1}^{t} \lambda_k \alpha_{ik} \gamma_{jk} + \varepsilon_{ij}$$

Where: $\bar{Y}_{ij}$ is the performance of genotype $i$ in environment $j$, $\mu$ is the grand mean, $\beta_j$ is the main effect of environment $j$, $k$ is the number of principal components (PC); $\lambda_k$ is singular value of the $k$th PC; and $\alpha_{ik}$ and $\gamma_{jk}$ are the scores of $i$th genotype and $j$th environment, respectively for $PC_k$; $\varepsilon_{ij}$ is the residual associated with genotype $i$ in environment $j$.

## Results and discussion

### Analysis of variance of root yield and yield-related traits

A combined analysis of variance revealed the presence of a highly significant difference (p<0.001) among tested genotypes for all the traits considered (Table 1). The genotype-by-environment interaction had a significant effect (p<0.001) on all traits except SPVD, demonstrating that different genotypes responded differently to test locations. Also, the presence of significant genotype-environment interaction is a clear indicator of genotypic differences and their sensitivity to various environments, which ultimately complicates the process of variety recommendation to wider areas [44]; hence, demanding further stability analysis to identify a stable genotype across locations as is essential suggested by [45]. The three-way interaction effect (genotype x location x year) was significant for all traits considered in this study, reflecting that the observed significant differences could be associated with genotypic differences, growing season, cultural management, and environmental conditions where the trial was conducted. The non-significant differences observed among evaluated genotypes for SPVD disease reaction may be related to some factors, such as unfavorable environmental conditions for disease development, including the absence of vectors that carry and spread virus diseases from plant to plant by sucking insects in the trial areas, this agrees with the earlier report [21].

**Table 1. Analysis of variance for various traits of ten OFSP genotypes evaluated at three locations over two years.**

| Source of variance | D.F | Mean squares | | | | | | |
|---|---|---|---|---|---|---|---|---|
| | | SPVD (1–9) | RL (cm) | RG (cm) | AGFW (t ha$^{-1}$) | TRYLD (t ha$^{-1}$) | DMC (%) | BCC (mg100g-1) |
| Rep (LxY) | 10 | 0.33$^{ns}$ | 67.15*** | 16.00** | 743.50*** | 6.17$^{ns}$ | 0.0002$^{ns}$ | 14.10 $^{ns}$ |
| Genotype (G) | 9 | 3.37*** | 126.47*** | 59.65*** | 592.64*** | 1807.89*** | 0.09*** | 26.21*** |
| Location (L) | 2 | 1.59$^{ns}$ | 1135.15*** | 2893.19*** | 25839.80*** | 1632.88*** | 0.06*** | 19.20** |
| Year (Y) | 1 | 10.58*** | 50.47** | 61.34*** | 870.58** | 13.68$^{ns}$ | 0.0003$^{ns}$ | 22.31** |
| GxL | 18 | 3.43*** | 28.86*** | 12.94** | 1075.73*** | 269.75*** | 0.06*** | 19.32** |
| GxY | 9 | 2.30** | 3.47$^{ns}$ | 15.02** | 152.00$^{ns}$ | 31.95*** | 0.0002$^{ns}$ | 21.30** |
| LxY | 2 | 17.35*** | 12.62$^{ns}$ | 19.51** | 15925.83*** | 462.35*** | 0.08*** | 23.00** |
| GxLxY | 18 | 1.57** | 5.93$^{ns}$ | 15.31** | 142.64$^{ns}$ | 76.95*** | 0.0002$^{ns}$ | 20.21** |
| Error | 108 | 0.73 | 5.41 | 6.00 | 175.87 | 5.17 | 0.0002 | 4.1 |

Where, 1–9 rating scale (1 = immune, 9 = Susceptible, hence 1–3 = resistant, 4–6 = medium and 7–9 = susceptible), D.F = degree of freedom, RL = Root length, RG = Root girth, AGFW = Above-ground fresh weight, TRYLD = Total root yield, DMC = Dry matter content, BCC = Beta-carotene content.

## Performance of genotypes for mean root yield across the evaluated environments

The root yield trait of genotypes evaluated varied significantly across test locations over the years (Table 2). The differences observed among the evaluated genotypes could be attributed to genetic variations inherent in each genotype, which resulted in varied responses when evaluated in different locations [16]. In year 1 (2021), the mean root yield of genotypes ranged from 6.4 t ha$^{-1}$ for genotype G9 (CORDNER-15-2) to 37.0 t ha$^{-1}$ for genotype G3 (13NC9350A-9-3). In year 2 (2022), for the same genotypes, G9 and G3, root yields ranged from 7.0 t ha$^{-1}$ to 35.5 t ha$^{-1}$, respectively, with an overall mean of 18.3 t ha$^{-1}$ in this study. In terms of each location performance, the highest root yield was recorded for G3, followed by G1 (MUSG014052-51-5) with values of 49.0 and 48.9 t ha$^{-1}$, respectively, from the Arbaminch

**Table 2. Root yield (t ha$^{-1}$) performance of ten OFSP genotypes evaluated across locations and over years.**

| Genotype code | Genotype name | 2021 | | | | 2022 | | | | Genotype by Environment | Yield advantage over standard check (%) |
|---|---|---|---|---|---|---|---|---|---|---|---|
| | | Hawassa | Koka | ArbaMinch | Mean | Hawassa | Koka | ArbaMinch | Mean | | |
| G1 | MUSG014052-51-5 | 14.0 | 13.5 | 48.9 | 25.5 | 13.7 | 8.7 | 32.6 | 18.3 | 21.9 | -14.5 |
| G2 | MUSG014001-3-7 | 10.7 | 15.9 | 30.6 | 19.0 | 12.9 | 17.4 | 20.4 | 16.9 | 18.0 | -29.7 |
| G3 | 13NC9350A-9-3 | 33.7 | 28.5 | 49.0 | 37.0 | 30.3 | 42.5 | 33.6 | 35.5 | 36.2 | 41.4 |
| G4 | CN1448-49-26-12 | 7.0 | 12.5 | 8.0 | 9.2 | 6.9 | 20.2 | 7.3 | 11.5 | 10.3 | -59.8 |
| G5 | CN1448-49-28-9 | 11.6 | 26.4 | 32.3 | 23.5 | 15.3 | 30.0 | 28.3 | 24.6 | 24.0 | -6.3 |
| G6 | 107031-18-5 | 7.7 | 8.5 | 4.7 | 7.0 | 8.1 | 6.5 | 7.4 | 7.3 | 7.1 | -72.3 |
| G7 | 105413–5 | 9.9 | 22.2 | 41.1 | 24.4 | 16.2 | 31.2 | 27.4 | 24.9 | 24.7 | -3.5 |
| G8 | 105413–13 | 6.5 | 10.0 | 4.2 | 6.9 | 7.2 | 7.7 | 7.0 | 7.3 | 7.1 | -72.3 |
| G9 | CORDNER-15-2 | 6.4 | 9.60 | 3.1 | 6.4 | 6.1 | 7.8 | 7.0 | 7.0 | 6.7 | -73.8 |
| G10 | Alamura (Check) | 13.2 | 23.4 | 41.3 | 25.9 | 14.8 | 36.3 | 24.6 | 25.2 | 25.6 | - |
| Mean | | 12.0 | 17.1 | 26.3 | 18.8 | 13.1 | 21.0 | 19.6 | 17.7 | 18.3 | |
| LSD (5%) | | 3.7 | 4.2 | 4.8 | 4.2 | 3.4 | 3.9 | 3.5 | 3.6 | 3.9 | |
| CV (%) | | 18.3 | 14.6 | 11 | 14.6 | 15.5 | 11.1 | 10.6 | 12.4 | 13.5 | |

location in year 1. In year 2, the highest root yield of 42.5 t ha$^{-1}$ for G3 was obtained from Koka location. In both years, genotype G3 was the highest performer over the stand-check variety, with a yield advantage of 41.4%. Two locations, namely Koka and Arbaminch, can be considered the best environment over Hawassa locations, as confirmed by the genotypes that perform the best in this study. Among the ten evaluated genotypes, genotype G3 produced the highest root yield (t ha$^{-1}$) in all environments (locations vs. years) (Table 2). The four geno-types G6 (107031-18-5), G8 (105413–13), and G9 (CORDNER-15-2) gave the lowest root yields, i.e., <10 t ha$^{-1}$, reflecting their poor performance under the evaluated environmental conditions. When evaluating and selecting sweetpotato genotypes for areas where irregular rainfall makes sustainable production of the existing crop varieties challenging, it is crucial to consider genotypes that excel at combined traits such as early maturation and high root yield for further recommendation [46].

## Mean performance of genotypes for root yield-related traits and SPVD reaction

A combined analysis of variance revealed the presence of a significant difference ($p < 0.05$) among genotypes in response to SPVD. The severity of symptoms for the evaluated genotypes ranged from 1.39 for G3 (13NC9350A-9-3) to 2.78 for G5 (CN1448-49-28-9), on a 1–9 rating scale, with an overall mean of 1.94 (Table 3). The results showed that the majority evaluated genotypes had a high level of tolerance/resistance to SPVD, with a low score below 3.0, falling within the range of resistance to SPVD severity [21,35]. The low level of disease severity among evaluated genotypes for SPVD, which indicates the presence of an unsuitable environment for the virus to evolve, as well as the lack of vectors or carriers that spread the virus from plant to plant near experimental sites [20,47]. While selecting and advancing sweetpotato genotypes for later stages, it is suggested to assess their sensitivity to SPVD reactions along with other desirable traits [20,21]. The significant differences ($p < 0.05$) were observed in root length (cm) trait. The root length varied from 7.98 cm to 15.10 cm for genotypes G8 (105413–13) and G3 (13NC9350A-9-3), respectively, with an overall mean of 11.78 cm (Table 3). Root

**Table 3. Mean performance of ten OFSP genotypes for various traits evaluated across locactions over years.**

| Code | Genotypes | SPVD (1–9 scale) | RL (cm) | RG (cm) | AGFW (t ha$^{-1}$) | HI (%) |
|------|-----------|------------------|---------|---------|---------------------|--------|
| G1 | MUSG014052-51-5 | 1.83 | 11.84 | 8.67 | 37.19 | 43.00 |
| G2 | MUSG014001-3-7 | 1.61 | 13.67 | 7.43 | 32.80 | 43.00 |
| G3 | 13NC9350A-9-3 | 1.39 | 15.10 | 9.03 | 27.74 | 61.00 |
| G4 | CN1448-49-26-12 | 1.72 | 10.51 | 6.47 | 24.50 | 32.00 |
| G5 | CN1448-49-28-9 | 2.78 | 13.76 | 9.87 | 41.24 | 49.00 |
| G6 | 107031-18-5 | 2.11 | 9.71 | 5.98 | 30.73 | 25.00 |
| G7 | 105413–5 | 2.55 | 12.06 | 10.41 | 34.31 | 50.00 |
| G8 | 105413–13 | 1.78 | 7.98 | 5.39 | 30.61 | 21.00 |
| G9 | CORDNER-15-2 | 1.66 | 8.09 | 6.51 | 26.47 | 20.00 |
| G10 | Alamura (Check) | 1.99 | 15.06 | 9.80 | 22.90 | 55.00 |
| Mean | | 1.94 | 11.78 | 7.96 | 30.85 | 40.00 |
| LSD (0.05) | | 0.78 | 3.34 | 4.32 | 8.67 | 9.00 |
| CV (%) | | 26.30 | 23.40 | 27.00 | 28.80 | 24.10 |

Where, 1–9 scale (1 = immune, 9 = Susceptible, hence 1–3 = resistant, 4–6 = medium and 7–9 = susceptible); SPVD = Sweetpotato virus disease, RL = Root length, RG = Root girth, AGFW = above ground fresh weight, HI = Harvest index.

girth trait ranged 5.39 cm to 10.41 cm, respectively for G8 (105413–13) and G7 (105413–5) with overall mean of 7.96 cm (Table 3). Root length and root girth traits are among the best traits in sweetpotato genotypes screening process as these two traits can be an indicator for earliness (early bulking). Preferred genotypes for market appeal include those with early bulking and roots of appropriate size, i.e., root length no longer than 15 cm, the smallest no less than 10 cm, and root-girth/diameter at least greater than 4 cm; this is roughly what most consumers stated as their ideal root sizes, and it is now incorporated into the target product traits profile.

Genotypes varied significantly for above-ground fresh weight (t ha$^{-1}$) in all environments, i.e., three locations vs. two years (Table 3). The highest mean yield of 41.24 t ha$^{-1}$ was recorded for G5 (CN1448-49-28-9) whereas the lowest mean yield of 24.50 t ha$^{-1}$ recorded for G4 (CN1448-49-26-12). Genotypes with combination of traits, such as high root yield and above-ground biomass yield (above-ground fresh weight) can be utilized for both human consumption and animal feed, particularly in areas under land constrain [48]. The harvest index (HI) performance of genotypes revealed significant variation. The genotype G3 (13NC9350A-9-3) had the highest mean value of 61% of HI, while the genotype G9 (CORDNER-15-2) had the lowest mean value of 20% of HI. A genotype with a high harvest index value is believed to have a high efficiency in storage root production in relationship to its biological yield [49]. The authors also explained that the harvest index signified the distribution of assimilation between economic and total plant biomass. Additionally, it was indicated that high yielding varieties have a higher harvest index than low yielding varieties in terms of root yield.

## Performance of OFSP genotypes for root dry matter, beta-carotene contents and flowering potential

The root dry matter contents (DMC) of the evaluated genotypes ranged from 24.0% to 31%, with an overall mean of 27.0% (Table 4). The highest mean percentage of 31% DMC was recorded for the check variety (Alamura), whereas the lowest mean of 24.0% DMC was recorded for genotype G6 (107031-18-5). The DMC, a key trait in the OFSP variety associated with the mealiness of boiled roots, can impact the wider acceptability of released varieties in

**Table 4. Mean root dry matter content, beta-carotene content and flowering ability of the evaluated sweetpotato genotypes.**

| Code | Genotypes | DMC (%) | BCC (gm/100g) | Flowering ability of evaluated genotypes |
|------|-----------|---------|---------------|------------------------------------------|
| G1 | MUSG014052-51-5 | 26.00 | 7.11 | Mostly not flowering or 2% |
| G2 | MUSG014001-3-7 | 25.00 | 3.65 | Mostly not flowering or 3% |
| G3 | 13NC9350A-9-3 | 30.00 | 5.50 | 95% flowering (profusely flowering) |
| G4 | CN1448-49-26-12 | 27.00 | 3.68 | Mostly not flowering or 3% |
| G5 | CN1448-49-28-9 | 27.00 | 3.67 | Mostly not flowering or 5% |
| G6 | 107031-18-5 | 24.00 | 3.72 | Mostly not flowering or 3% |
| G7 | 105413–5 | 30.00 | 3.75 | Mostly not flowering or 5% |
| G8 | 105413–13 | 25.00 | 3.50 | Mostly not flowering or 3% |
| G9 | CORDNER-15-2 | 25.00 | 3.48 | Mostly not flowering or 3% |
| G10 | Alamura (Check) | 31.00 | 7.46 | Mostly not flowering or 10% |
| Mean | | 27.00 | 4.60 | |
| LSD (0.05) | | 2.00 | 0.55 | |
| CV (%) | | 15.50 | 7.20 | |

Where, DMC = Root dry matter content, BCC = Beta-carotene content = estimated on a fresh weight basis based on based on colour chart [38].

areas where this crop is widely consumed, such as East African countries [16]. According to Makunde et al. (2017) and Fuglie (2007), in order to meet consumer preferences in the humid tropics, an OFSP dry matter level categorized as medium between 24% and 28% and a high dry matter content greater than 28% are required [17,18]. Accordingly, in this study, three genotypes, such as G3 (13NC9350A-9-3), G7 (105413–5), and the check variety Alamura, had DMCs of 30%, 30%, and 31%, respectively, and can be considered the best genotypes for dry matter content (Table 4).

Another vital nutritional pigment found in OFSP variety is beta-carotene (BCC), which distinguishes OFSP varieties from white-fleshed ones and is a precursor to vitamin A, one of the most important vitamins for human health. The performance of the evaluated genotypes significantly differed at $p < 0.05$ in test environments (Table 4). The BCC for genotypes was estimated based on a root flesh color chart [38]. As explained in that colour chart standards, the deeper orange root flesh colour of the OFSP genotype indicates a higher beta-carotene concentration. More importantly, it was defined that the ranges for the most favorable genotypes may be defined as orange or intermediate orange, ranging from 5.08 to 8.36 mg/100 g on a fresh weight basis [38]. In this study, check variety (Alamura) had the highest mean value of 7.46 mg/100 g, whereas genotype G9 (CORDNER-15-2) had the lowest mean value of 3.48 mg/100 g. In this study, three genotypes, G1 (MUSG014052-51-5), G3 (13NC9350A-9-3), and G10 (Alamura), showed a desirable BCC range, which is equal to ranges specified as acceptable on a fresh weight basis [38]. A preferred genotype is one with a relatively high beta-carotene content combined with higher root yields; thus, it is essential to select a suitable genotype for production as a systematic diet in areas where there are nutritional challenges [50].

In terms of flowering ability, most of the evaluated genotypes seldom produced flowers under test environments; however, genotypes with higher flowering potential are desired since they may be used as parents for future hybridization purposes. In this study, genotype G3 performed very well for this trait, i.e., flowing profusely enough that it can be used as a parent for future cross-breeding/hybridization purposes upon further study of the traits of interest (Table 4).

## Genotype by environment interaction and stability analysis of genotypes using a GGE-biplot

Fig 1 depicts the relative performance of ten OFSP genotypes for root yields across locations based on GGE-biplot analysis. In this study, the first two principal components (PC), PC1 and PC2, were able to explain 84.14% and 10.88% of variation, respectively, with an overall variation of 95.02%, indicating that the first two principal components (PC1 and PC2) adequately explained the variations, which meets the criteria for using a GGE-biplot (Fig 1). Following an interpretation of their respective values (scores) from a GGE biplot analysis, large positive PC scores for a specific genotype indicate that it is considered a higher average value, whereas those with large negative PC scores indicate that they are considered to be a lower score value [43,44]. Also, it was explained as those genotypes closer to the origin of the biplot can be considered stable and those farther away as unstable [51]. The authors also explained that genotypes closer to specific environments are more adaptive to that environment. The performance of a genotype is significantly influenced by the environment in which it is grown [45]. In contrast to widely adapted stable and non-responsive varieties, environment-specific adapted varieties, i.e., unstable and responsive varieties, have a tendency to respond to environmental changes [52]. This confirms the earlier claim that if all environmental markers fall into one sector, it means that a single cultivar has the highest yield in all environments (across locations and over years). In contrast, if environmental markers fall into separate sectors, it suggests that different cultivars excel in different environments where they were evaluated.

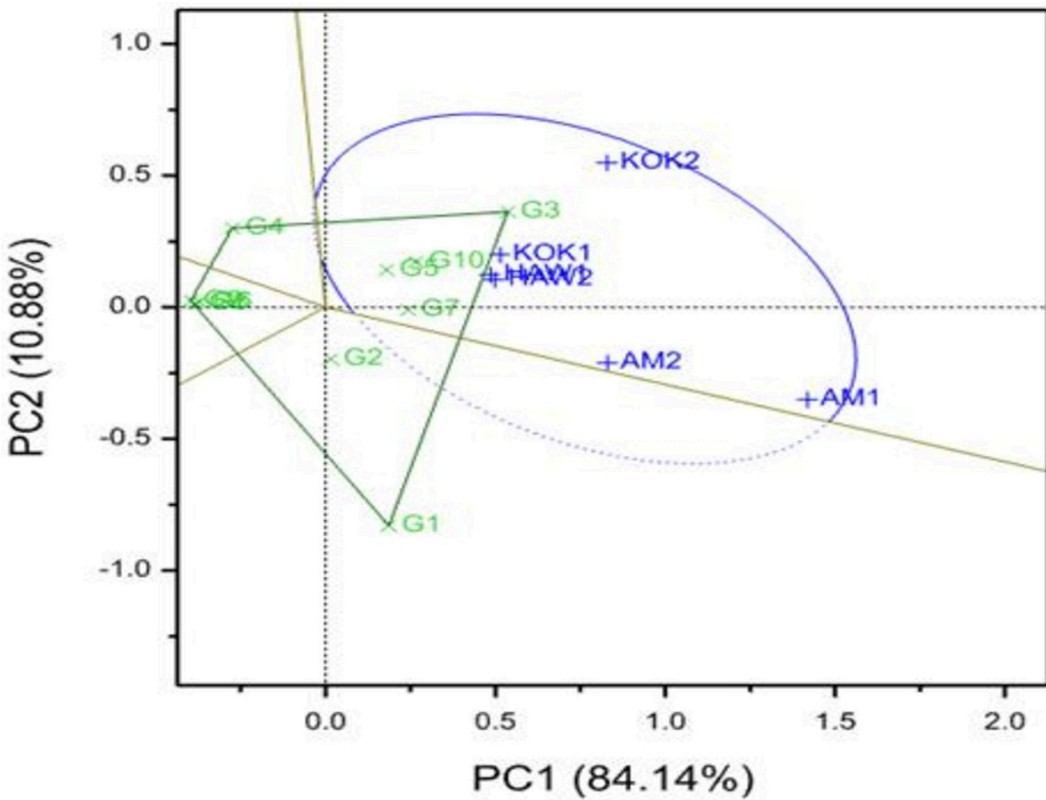

**Fig 1. Showing the "which-won-where" pattern of tested genotypes for total root yield performance across locations over years b*ased o*n *ge*notype plus genotype-by-environment (GGE) biplot.** Where, HAW1 = Hawassa year 1; HAW2 = Hawassa year 2; Kok1 = Koka year 1; Kok2 = Koka year 2; AM 1 = Arba Minch year 1; AM2 = Arba Minch year 2.

Based on GGE-biplot analysis, five genotypes such as G3 (13NC9350A-9-3), G10 (Alamura), G7 (105413–5), G1 (MUSG014052-51-5), and G5 (CN1448-49-28-9), gave the relatively highest root yields, which were above average yields. While four genotypes such as G6 (107031-18-5), G8 (105413–13), G9 (CORDNER-15-2), and G4 (CN1448-49-26-12) had the lowest average root yields. Genotypes G5 (CN1448-49-28-9), G2 (MUSG014001-3-7), and G7 (105413–5) had low PC2 values close to zero, indicating that they were relatively stable. Furthermore, Yan et al. (2007), explained that stable genotypes are those with a small vector distance from the center of the biplot, as determined by a GGE biplot analysis [51]. Out of the ten genotypes that were evaluated, G7 (105413–5) is low yielding but more stable than G3 (13NC9350A-9-3) and G10 (Alamura) (Fig 1). The vertex genotypes at the polygons such as G1, G3, G4, G9, and G8 were performed either well or poorly, which is in line with earlier reports that explained the results obtained from GGE biplot analysis for sweetpotato genotypes tested in diverse environments [22,31].

Breeding and evaluating crop varieties for specific areas may be challenging tasks because of low resources investment for breeding operations in the variety development process; thus, selection of a relatively stable genotype with better yielding potential across diverse environments is considered as the simple, and cost-effective approaches to identify relatively best genotypes for recommendation. In this study, genotype designated as G3 was performed well almost in all environments (locations vs. years), i.e., it was found at the apex of the sector, and it is a high yielder with broad adaptation, which can be considered as the best genotype to identify and recommend for wider production. According to Yan et al. (2007), the length of

the environmental vector shows the magnitude of the influence of genotypic factors, the environments, and their interactions [51].

In terms of evaluated environments, environment designated as AM1 and AM2 (Arbaminch location in year 1 and year 2) exhibited relatively large PC1 scores, they were considered as better discriminated among genotypes for root yield. While environments designated as HAW1 and HAW2 (Hawassa location in year 1 and year 2) had PC2 scores that were close to zero, however, they were considered poor performance environments for root yield during this study period (Fig 1). The findings of this study are consistent with a previous report by Gurmu (2017), who conducted a sweetpotato field trial at six locations and suggested stable and adaptable genotypes using a GGE-biplot analysis [29].

The mean yield versus stability view biplot was computed and depicted based on ten OFSP genotypes evaluated in six diverse environments (three locations and two years) (Fig 2). The average environment coordinate (AEC) abscissa (the vertical line that passes through the origin of the biplot) divides genotypes into two groups: high yielders that yield more than the overall mean (right hand side) and lower mean yielders that yield less than the overall mean (left hand side). Based on this, the genotypes G7, G3, G5, and G10 produced higher root yields than the overall mean of genotypes. While genotypes G6, G8, and G9 had the lowest root yield across all tested environments (Fig 2), a vector is used to connect each genotype to the average

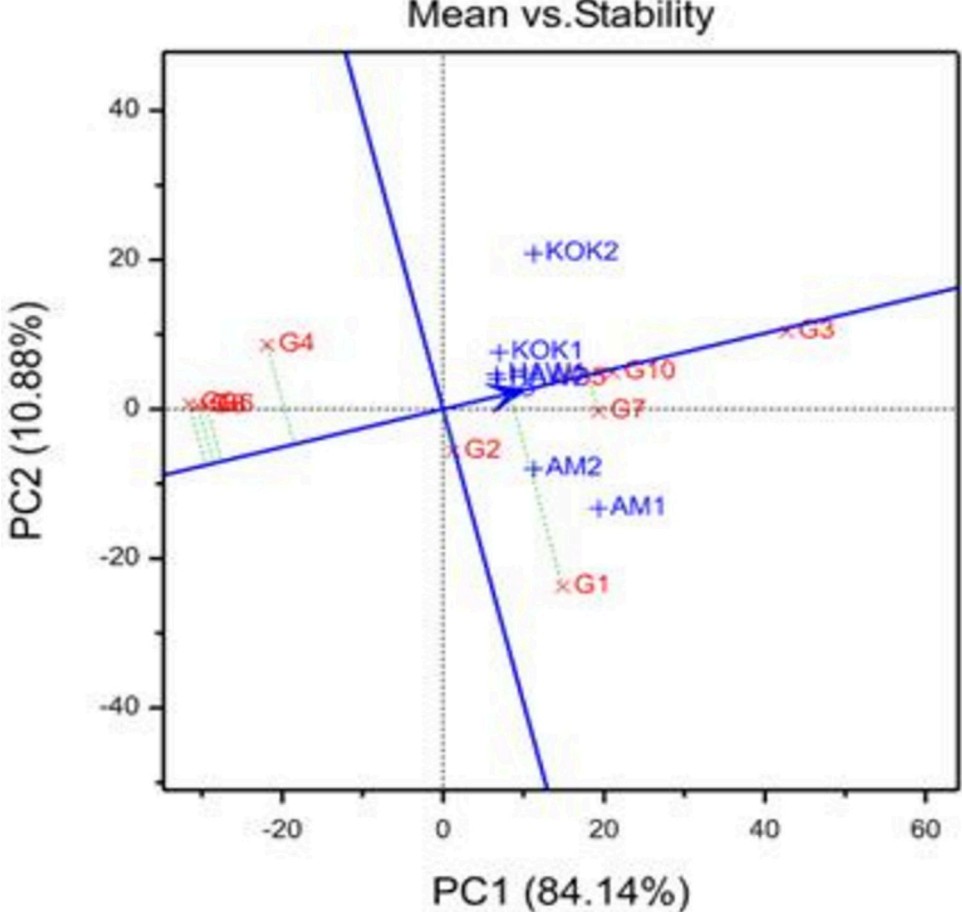

**Fig 2. The mean performance and stability of 10 evaluated orange-fleshed sweetpotato genotypes in six environments.**

environmental axis, demonstrating genotype stability. The genotypes with a short vector are stable, whereas those with long vectors are unstable across test environments. In this context, G3, G7, and G10 were considered the most stable. Among the stable genotypes, G3 had the highest yield in all environments (across locations and years) and did not demonstrate specific adaptability to only one or two environments, reflecting that it could be a suitable genotype across wider sweetpotato production domains [29–31].

**A GGE-biplot-based evaluation of test environments for representativeness and discriminating ability.** According to the explanation provided for the GGE-biplot analysis by [41–43,51], environments with high PC1 scores are considered to be better at distinguishing between genotypes, whereas those with low PC2 values are more representative of the average. The authors also proposed using a biplot to determine the representativeness of a test site, using the average environment as a reference. The average environment coordinate (AEC) abscissa was determined by averaging PC1 and PC2 values across all environments. The proximal angle with the AEC defines the representativeness of an individual location; the smaller the angle between the location vector and the AEC, the more representative the tested site. Based on the GGE explanations, the two environments in this study, designated as HAW1 (Hawassa year 1) and HAW2 (Hawassa year 2), were the most representative in terms of root yields of the tested genotypes, including KOK1 (Koka year 1), whereas KOK2 (Koka year 2), AM1 (Arbaminch year 1), and AM2 (Arbaminch year 2) were the least representative in terms of root yields of the evaluated genotypes [Fig 3]. Two environments designated as AM1 and AM2 are better distinguished between genotypes for root yield, reflecting that they can be better test environments than the others. The current findings of this study are consistent with earlier findings on sweetpotato genotype evaluations and recommendations of sweetpotato cultivars suitable for varied agro-ecological conditions in Ethiopia [29,31].

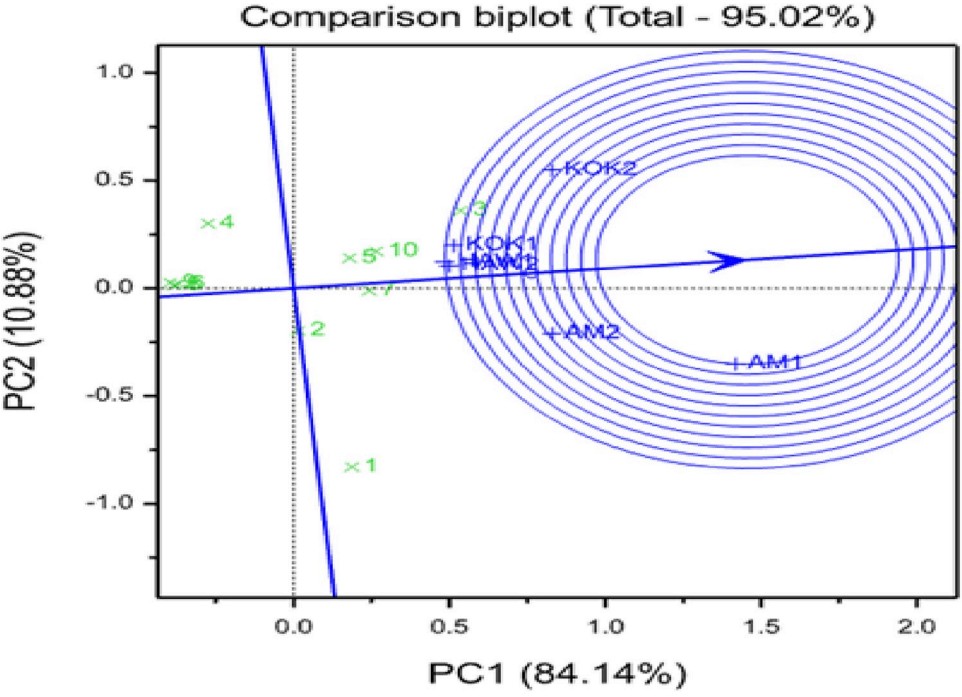

**Fig 3. The vector view of GGE biplot of 10 OFSP genotypes evaluated across locations over years.**

## Conclusion

The development of orange-fleshed sweetpotatoes that mature early is crucial for lowland cultivation, where erratic/irregular rainfall poses a major problem for sustainable production. This type of varieties is regarded as more important since they can achieve maturity sooner, allowing harvesting even in short growing seasons. Based on this fact, sweetpotato genotypes with the earliness attribute have the potential to greatly boost production and productivity by ensuring resilience to present climate shocks. The current study was designed to develop, evaluate and select suitable genotypes in terms of earliness and high root yields under diverse environments. As a result, out of ten genotypes evaluated in multi-environment trial over two years, genotype G3 (13NC9350A-9-3), outperformed all genotypes in all environments (across locations and years). The graphical analysis of GGE biplot was also revealed that G3 was the vertex genotype of the polygon that performed the best. Overall, this genotype demonstrated consistent performance across locations, including resistance to SPVD severity, had optimum level of beta-carotene and dry matter contents, and earliness that it can matures within 3–4 months. Also, G3 can be a good genotype for improving crop diversity since it provides an opportunity while leaving space for other crops because of its early maturation. In general, based on the overall performance, G3 is recommended for verification and release for wider production in mid to lowland areas in Ethiopia.

## Supporting information

**S1 Data. Weather information for the testing locations in 2021 and 2022.**
(XLSX)

**S1 Table. Description of experimental materials used for the study.**
(DOC)

## Author Contributions

**Conceptualization:** Bililign Mekonnen, Fekadu Gurmu.

**Data curation:** Bililign Mekonnen, Fekadu Gurmu.

**Formal analysis:** Bililign Mekonnen.

**Funding acquisition:** Bililign Mekonnen.

**Investigation:** Bililign Mekonnen.

**Methodology:** Bililign Mekonnen.

**Project administration:** Bililign Mekonnen.

**Resources:** Bililign Mekonnen, Fekadu Gurmu.

**Software:** Bililign Mekonnen.

**Supervision:** Bililign Mekonnen, Fekadu Gurmu.

**Validation:** Fekadu Gurmu.

**Visualization:** Bililign Mekonnen.

**Writing – original draft:** Bililign Mekonnen.

**Writing – review & editing:** Bililign Mekonnen, Fekadu Gurmu.

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
