## [Decision Letter · Decision Letter 0]

30 Jan 2024

PONE-D-23-35204Evaluation of advanced early maturing orange-fleshed sweetpotato (Ipomoea batatas [L].Lam) genotypes in selected areas across EthiopiaPLOS ONE

Dear Dr. WUbe,

Thank you for submitting your manuscript to PLOS ONE. After careful consideration, we feel that it has merit but does not fully meet PLOS ONE’s publication criteria as it currently stands. Therefore, we invite you to submit a revised version of the manuscript that addresses the points raised during the review process.

We look forward to receiving your revised manuscript.

Kind regards,

Kun Lu, Ph.D.

Academic Editor

PLOS ONE

A clean copy of the edited manuscript (uploaded as the new *manuscript* file)".

4. We notice that your supplementary figures are uploaded with the file type 'Figure'. Please amend the file type to 'Supporting Information'. Please ensure that each Supporting Information file has a legend listed in the manuscript after the references list.

Reviewers' comments:

Reviewer's Responses to Questions

**Comments to the Author**

1. Is the manuscript technically sound, and do the data support the conclusions?

Reviewer #1: Yes

Reviewer #2: Yes

2. Has the statistical analysis been performed appropriately and rigorously? 

Reviewer #1: No

Reviewer #2: Yes

3. Have the authors made all data underlying the findings in their manuscript fully available?

Reviewer #1: Yes

Reviewer #2: No

4. Is the manuscript presented in an intelligible fashion and written in standard English?

Reviewer #1: Yes

Reviewer #2: No

5. Review Comments to the Author

Reviewer #1: Good Research, however, a stability analysis either AMMI or other form of stability analysis should be done before the conclusion on G3 can be made. This is because of the genotype by environment interaction which is significant. I suggest you run a stability analysis.

Reviewer #2: 1. Clear Objective: The manuscript clearly outlines its purpose, which is to assess the performance of advanced early maturing orange-fleshed sweetpotato genotypes in Ethiopia.

2. Relevance: The research addresses an important agricultural and nutritional concern in Ethiopia, as orange-fleshed sweetpotatoes are a valuable source of nutrition. The study's findings could potentially contribute to addressing food security and nutritional challenges.

3. Geographical Diversity: The inclusion of multiple regions in Ethiopia for the evaluation provides a comprehensive understanding of how these genotypes perform across diverse climates and soil conditions, making the study more applicable to various areas.

4. Focus on Early Maturity: The emphasis on early maturation is significant, as it aligns with the need for crops that can be harvested sooner, potentially increasing overall agricultural productivity.

5. The materials and methods section is well written and the statistical models well described.

6.Consistenttly refer to the genotypes by the same names- the authors are using the genotype names like 13NC9350A-9-3,CN1448-49-26-12, 107031-18-5, 105413-5, 105413-13, CORDNER-15-2 interchangeably with codes like G1,G3, G2 etc. This interrupts the flow of the reader as they have to now figure out which on you are refereeing to.

7. Although the manuscript shows data in tables, figures and supplementary materials, unfortunately I do not see a data availability statement in the manuscript

6. PLOS authors have the option to publish the peer review history of their article (what does this mean?). If published, this will include your full peer review and any attached files.

Reviewer #1: No

Reviewer #2: **Yes: **Ssali Reuben Tendo

---

## [Author Response · Author response to Decision Letter 0]

28 Mar 2024

With regards to the suggestion to "consistently refer to the genotypes by the same names," the suggested correction has been made.

The data availability declaration was mentioned both in the reply letter and during submission.

To satisfy the PloS ONE requirements, the original manuscript underwent extensive revisions with regard to scientific soundness.

---

## [Decision Letter · Decision Letter 1]

23 Jun 2024

PONE-D-23-35204R1Evaluation of the performance and stability of early-maturing orange-fleshed sweetpotato genotypes in selected moisture stress areas in EthiopiaPLOS ONE

Dear Dr. Wube,

Thank you for submitting your manuscript to PLOS ONE. After careful consideration, we feel that it has merit but does not fully meet PLOS ONE’s publication criteria as it currently stands. Therefore, we invite you to submit a revised version of the manuscript that addresses the points raised during the review process.

We look forward to receiving your revised manuscript.

Kind regards,

Jiban Shrestha

Academic Editor

PLOS ONE

Additional Editor Comments:

The authors need to include all suggestions given by reviewers.

Reviewers' comments:

Reviewer's Responses to Questions

**Comments to the Author**

1. If the authors have adequately addressed your comments raised in a previous round of review and you feel that this manuscript is now acceptable for publication, you may indicate that here to bypass the “Comments to the Author” section, enter your conflict of interest statement in the “Confidential to Editor” section, and submit your "Accept" recommendation.

Reviewer #1: All comments have been addressed

Reviewer #3: (No Response)

Reviewer #4: All comments have been addressed

2. Is the manuscript technically sound, and do the data support the conclusions?

Reviewer #1: Yes

Reviewer #3: (No Response)

Reviewer #4: Partly

3. Has the statistical analysis been performed appropriately and rigorously? 

Reviewer #1: (No Response)

Reviewer #3: (No Response)

Reviewer #4: Yes

4. Have the authors made all data underlying the findings in their manuscript fully available?

Reviewer #1: Yes

Reviewer #3: (No Response)

Reviewer #4: No

5. Is the manuscript presented in an intelligible fashion and written in standard English?

Reviewer #1: Yes

Reviewer #3: (No Response)

Reviewer #4: No

6. Review Comments to the Author

Reviewer #1: The authors have effected the necessary corrections, the manuscript should therefore be accepted for publication

Reviewer #3: The article lacks detailed discussion in terms of comparison with past related studies done by other researchers. Hence they should improve discussion of their results.

they should improve their English.

Reviewer #4: Following points needs clarification and are as follows-

1. The intricacy of the article cannot be adequately conveyed by the abstract and introduction. The Sweetpotato Virus Disease (SPVD) was graded, yet there is a lack of information in the introduction. The introduction has not addressed root qualities, their function, and a rational debate of drought tolerance to enhance comprehension. The term "drought" is frequently used in the article because it is a challenging parameter to measure, usually using physical instruments and techniques that require specialized knowledge.

2. The current study used nine genotypes, which is a smaller number. It's unclear which check is being used. Check-related details ought to have been expanded upon.

3. Instead of using "drought and moisture stress condition," I advise the authors to use the phrase irregular rainfall or inadequate rainfall in entire manuscript.

The goal of the study, which I believe may not be achieved by carrying out the research, is to find early-maturing genotypes or varieties that may mature early and avoid the consequences of low moisture stress brought on by protracted periods of drought.

4. It is necessary to include more details regarding the genotypes of sweet potatoes that mature early and late in the article as well as in introduction.

5. There may be a need to improve the disease screening procedures because non-significant differences in SPVD were observed across locations.

6. While the genotype that was identified may be useful in the local context, it may not be of interest to breeders of sweet potatoes internationally.

7. PLOS authors have the option to publish the peer review history of their article (what does this mean?). If published, this will include your full peer review and any attached files.

Reviewer #1: No

Reviewer #3: **Yes: **Aleck Kondwakwenda

Reviewer #4: **Yes: **MANGESH Y. DUDHE Senior Scientist (Plant Breeding) ICAR-IIOR Rajendranagar Hyderabad 500030 India

---

## [Author Response · Author response to Decision Letter 1]

29 Jul 2024

All points raised by reviewers were addressed in the main document and also, indicated in rebuttal letter

---

## [Decision Letter · Decision Letter 2]

20 Aug 2024

PONE-D-23-35204R2Evaluation of the performance and stability of early-maturing orange-fleshed sweetpotato genotypes in selected areas in EthiopiaPLOS ONE

Dear Dr. Wube,

Thank you for submitting your manuscript to PLOS ONE. After careful consideration, we feel that it has merit but does not fully meet PLOS ONE’s publication criteria as it currently stands. Therefore, we invite you to submit a revised version of the manuscript that addresses the points raised during the review process.

We look forward to receiving your revised manuscript.

Kind regards,

Jiban Shrestha

Academic Editor

PLOS ONE

Journal Requirements:

Additional Editor Comments:

The authors need to include all suggestions given by reviewers.

Reviewers' comments:

Reviewer's Responses to Questions

**Comments to the Author**

1. If the authors have adequately addressed your comments raised in a previous round of review and you feel that this manuscript is now acceptable for publication, you may indicate that here to bypass the “Comments to the Author” section, enter your conflict of interest statement in the “Confidential to Editor” section, and submit your "Accept" recommendation.

Reviewer #5: All comments have been addressed

Reviewer #6: All comments have been addressed

2. Is the manuscript technically sound, and do the data support the conclusions?

Reviewer #5: Partly

Reviewer #6: Yes

3. Has the statistical analysis been performed appropriately and rigorously? 

Reviewer #5: Yes

Reviewer #6: Yes

4. Have the authors made all data underlying the findings in their manuscript fully available?

Reviewer #5: Yes

Reviewer #6: Yes

5. Is the manuscript presented in an intelligible fashion and written in standard English?

Reviewer #5: Yes

Reviewer #6: Yes

6. Review Comments to the Author

Reviewer #5: 1. Please check the interpretation on the GGE Biplot, Mean vs Stability. Address the comments in the section as indicated in the manuscript.

2. While quoting references in the text, at several places text abruptly ends in brackets indicating reference number for example : determined using the procedures suggested by [36, 37]. Please check whether it is the standard way of quoting as per the journal.

3. Other minor comments in the manuscript may be addressed

Reviewer #6: I have read all the previous reviewers comments, and same has been attempted by author. The minor correction are required as suggested in track changes before publication

7. PLOS authors have the option to publish the peer review history of their article (what does this mean?). If published, this will include your full peer review and any attached files.

Reviewer #5: **Yes: **P. Sanjana Reddy

Reviewer #6: **Yes: **Rajkumar Ramteke

---

## [Author Response · Author response to Decision Letter 2]

23 Aug 2024

All suggestions and comments given have been addressed and indicated in track changer in the main document.

---

## [Editor Report · Decision Letter 3]

28 Aug 2024

Evaluation of the performance and stability of early-maturing orange-fleshed sweetpotato genotypes in selected areas in Ethiopia

PONE-D-23-35204R3

Dear Dr. Wube,

We’re pleased to inform you that your manuscript has been judged scientifically suitable for publication and will be formally accepted for publication once it meets all outstanding technical requirements.

Kind regards,

Jiban Shrestha

Academic Editor

PLOS ONE

Additional Editor Comments (optional):

The authors need to check all references, tables/figures carefully.
---

## [Editor Report · Acceptance letter]

16 Oct 2024

PONE-D-23-35204R3 

PLOS ONE

Dear Dr. Mekonnen, 

I'm pleased to inform you that your manuscript has been deemed suitable for publication in PLOS ONE. Congratulations! Your manuscript is now being handed over to our production team.

Kind regards, 

on behalf of

Dr. Jiban Shrestha 

Academic Editor

PLOS ONE